# Learn to Follow: Lifelong Multi-agent Pathfinding with Decentralized Replanning

## Abstract

Multi-agent Pathfinding (MAPF) problem generally asks to find a set of conflict-free paths for a set of agents confined to a graph. In conventional MAPF scenarios, the graph and the agents' start and goal locations are known in advance. Thus, a centralized planning algorithm can be utilized to generate a solution. In this work, we investigate the decentralized MAPF setting, in which the agents can not share the information and must independently navigate toward their goals without knowing the other agents' goals or paths. We focus on the lifelong variant of MAPF, which involves continuously assigning new goals to the agents upon arrival to the previous ones. To address this complex problem, we propose a method that integrates two complementary approaches: planning with heuristic search and reinforcement learning (RL) through policy optimization. Planning is utilized to maintain an individual path, while RL is employed to discover the collision avoidance policies that effectively guide an agent along the path. This decomposition and intrinsic motivation specific for multi-agent scenarios allows leveraging replanning with learnable policies. We evaluate our method on a wide range of setups and compare it to the state-of-the-art competitors (both learnable and search-based). The results show that our method consistently outperforms the competitors in challenging setups when the number of agents is high.

## 1  Introduction

Multi-agent pathfinding (MAPF) [1] is a challenging problem that gets increasing attention recently. It is often studied in the AI community with the following assumptions. The agents are confined to a graph, and at each time step an agent can either move to an adjacent vertex or stay at the current one. A central controller possesses information about the graph and the agents' start and goal locations. This unit is in charge of constructing a set of conflict-free plans for all the agents. Thus, a typical setting for MAPF can be attributed as *centralized* and *fully observable*.

In many real-world domains, however, the central controller does not exist, or, even if it does, it may not possess full information about the environment. For example, consider a fleet of service robots delivering some items in a human-shared environment, e.g., the robots delivering drugs in the hospital. Each of these robots is likely to have access to the global map of the environment (e.g., the floor plan), possibly refined through the robot's sensors. However, the connection to the central controller may not be consistent. Thus, the latter may not have accurate data on the robots' locations and, consequently, cannot provide valid MAPF solutions. In such scenarios, *decentralized approaches* to the MAPF problems, when the robots themselves have to decide their future paths, are essential. Moreover, decentralized approaches may be preferable due to the poor scalability of the centralized ones. In this work, we aim to develop such an efficient decentralized approach.

It is natural to frame the decentralized MAPF as a sequential decision-making problem where at each time step, each agent must choose and execute an action that will advance it to the goal and, at the same time, will not disallow other agents to reach their goals as well. The result of solving this problem is a policy that, at each moment, tells which action to execute. To form such a policy, learnable methods are commonly used, for example, reinforcement learning (RL), which is especially beneficial in tasks with incomplete information [2, 3, 4]. However, even state-of-the-art model-free RL methods generally cannot efficiently solve long-horizon problems with the involved casual structure [5, 6], and they are often inferior to the seach-based methods when solving problems with hard combinatorial structure.

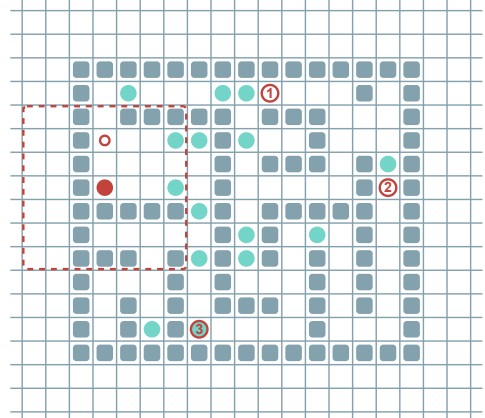

Figure 1: An example of a decentralized LMAPF instance is depicted below. Each agent is represented by a filled circle. The red agent has visibility limited to the positions of other agents within its field of view, indicated by a dotted red line. The red circles with numbers represent the goals that the agent needs to reach. The next goal is revealed only after the previous one is achieved. The small red circle indicates the subgoal the agent needs to accomplish to progress towards its goal.

The additional challenges that make the MAPF problems challenging for RL are as follows. First, we want the policy to be highly generalizable to previously unseen environments, which may differ significantly in scale and topology from the ones used during the learning stage. In MAPF, our primary interest lies not in how well the agents learn to behave in the environment(s) used for training, but rather how well they perform in any arbitrary (even out-of-the-distribution) environment. Second, MAPF problems are naturally dependent on the goal locations of the agents, meaning that even in the same environment (map), the goals may vary significantly. Finally, effectively training in a complex observation and action spaces poses challenges even for state-of-the-art multi-agent reinforcement learning (MARL) methods.

To this end, in this work we suggest not to solve the MAPF problem directly by RL but rather to decompose it into a series of sub-tasks utilizing heuristic search algorithms and then solve these sub-tasks efficiently with a learnable policy, that is obtained through the decentralized training. The general pipeline of our solution is the following. Each agent plans an individual path to its goal by the conventional heuristic search algorithm without considering the other agents (we also introduce an additional technique to penalize paths that are likely to cause deadlocks). Then a waypoint on this path is chosen in some vicinity of the agent, which becomes its local goal. To reach it, a learnable policy is utilized, which takes both static obstacles and the locally observable agents into account. Once a waypoint is reached, or the agent goes too far away from it the cycle repeats.

Empirically we compare our method, which we name FOLLOWER, to a range of both learnable and non-learnable state-of-the-art competitors and show that it *i*) consistently outperforms the competitors when the number of agents is high; *ii*) better generalizes to unseen environments compared to other learnable solvers; *iii*) may outperform a centralized search-based solver in certain setups.

## 2 Related Works

**Lifelong MAPF** LMAPF is an extension of MAPF when the agents are assigned new goals upon reaching their current ones. Similarly, in (online) multi-agent pickup and delivery (MAPD), agents are continuously assigned tasks, comprising two locations that the agent has to visit in a strict order – pickup location and delivery location. Typically, the assignment problem is not considered in LMAPF/MAPD. However, there exist works that include the assignment task into the problem, see [7, 8] for example.

In [9], several variants to tackle MAPD were proposed differing in the amount of data the agents share. Yet, even the decoupled (as attributed by the authors) algorithms based on Token Swapping rely on global information, i.e., the one provided by the central unit. An enhanced Token Swapping

variant that considers kinematic constraints was introduced in [10]. In [11], an efficient rule-based re-planning approach to solve MAPF was introduced that is naturally capable of solving LMAPD/MAPD problems. It did not rely on the several restrictive assumptions of Token Swapping and was empirically shown to outperform the latter.

Finally, one of the most recent and effective LMAPF solvers is the RHCR algorithm presented in [12]. It relies on the idea of bounded planning, i.e., constructing not a complete plan but rather its initial part. RHCR is a centralized solver that relies on the full knowledge of the agents' locations, their current paths, goals, etc. In this work we empirically compare with RHCR and show that our method is superior when the number of agents is high.

**Decentralized MAPF**    This setting entails that the paths/actions of the agents are decided not by a central unit but by the agents themselves. Numerous approaches, especially the ones tailored to the robotics applications, boil this problem down to reactive control, see [13, 14, 15] for example. These methods, however, are often prone to deadlocks. Several MAPF algorithms can also be implemented in a decentralized manner. For example, in [16] MAPP algorithm was introduced that relies on the individual pathfinding for each agent and a set of rules to determine priorities and choose actions to avoid conflicts when they happen along the paths. In [11] PIBT algorithm was introduced in which the agents also pick their actions individually (at each time step) based on specific rules. In general, most rule-based MAPF solvers, like [17], can be implemented in such a way that each agent decides its actions. However, in this case, the implicit assumption is that the agents can communicate to share the relevant information (or that they have access to the global MAPF-related data). In contrast, our work assumes that the agents are unable to communicate with one another or a central unit, which significantly increases the complexity of the problem.

**Learnable MAPF**    This direction has been getting increased attention recently. In [18], a seminal PRIMAL method that utilized reinforcement learning and imitation learning to solve MAPF in a decentralized fashion was introduced. Later in [19], it was also tailored to solve LMAPF. The new version got the name PRIMAL2. Since that, numerous learning-based MAPF solvers emerged, and it became common to compare against PRIMAL/PRIMAL2 (we also compare with it in our work). For example, in [20], another learning-based approach was proposed, tailored explicitly to agents with a non-trivial dynamic model, such as quadrotors. In [21] DHC method that utilized the agents' communications to solve decentralized MAPF efficiently was described. Another communication-based learnable approach, PICO, was presented in [22]. Overall, currently, a wide range of learnable decentralized MAPF solvers exist. However, to the best of our knowledge, they all rely on the communication between the agents or on access to the global MAPF-related data (like in PRIMAL, where each agent knows the goal locations of the others). We lift these assumptions in this work.

**MARL**    A separate direction in RL can be distinguished that specifically considers the multi-agent setting (MARL) [23]. Mainly these approaches consider game environments (like Starcraft [24]) in which pathfinding is not of the primary importance. However, several MARL methods, such as QMIX [3], MAPPO [25], have been adapted specifically for the MAPF task [26]. However they rely on the information sharing between agents.

Much attention is paid to multi-agent learnable methods in robotics [27]. Often, the value-based approaches are used to control small groups of agents on simple maps lile in [28] where a group of 4 agents is considered. In [29], a combination of Particle Swarm Optimization and Q-Learning controlling up to 100 agents is used. In [30], the model-based DynaQ method is used to learn agents in the knowledge exchange mode. Some works [31, 32] use value-based approaches with prior knowledge of how to interact with other agents. In [33] (MAPPER) an evolutionary reinforcement learning was used for MAPF task. This work also uses a global planner to determine sub-goals in learning one agent. In multi-agent mode, agents using ineffective polices are eliminated and only successful agents continue to be trained.

## 3    Background

**Multi-agent Pathfinding**    In (Classical) Multi-agent pathfinding [1], the timeline is discretized to the time steps, $T = 0, 1, 2, ...$ and the workspace, where $K$ agents operate, is discretized to a graph $G = (V, E)$, whose vertices correspond to the locations and the edges to the transitions between

these locations. $K$ start and goal vertices are given and each agent $i$ has to reach its goal $g_i \in V$ from the start $s_i \in V$. At each time step, an agent can either stay in its current vertex or move to an adjacent one. An individual plan for an agent $p_i$[1] is a sequence of actions that transfers it between two designated vertices. The plan's cost is the time step when the agent reaches the goal.

The MAPF problem asks to find a set of $K$ plans s.t. each agent reaches the goal without colliding with other agents. Formally, two collisions are usually distinguished: vertex collision, when the agents occupy the same vertex at the same time step, and edge collision, when the agents use the same edge at the same time step.

*Lifelong MAPF* (LMAPF) is a variant of MAPF where immediately after an agent reaches its goal, it is assigned to another one (via an external assignment procedure) and has to continue its operation. Thus, LMAPF generally asks to find not a fixed set of $K$ plans but rather to *i*) find a set of $K$ initial plans and *ii*) update each agent's plan when it reaches the current goal and receives a new one. In extreme cases, when some goal is reached at each step, the plans' updates are needed constantly (i.e., at each time step).

**The Considered Decentralized LMAPF Problem**   Consider a set of agents operating in the shared environment, represented as a graph $G = (V, E)$. The timeline is discretized to the time steps $T = 0, 1, ..., T_{max}$, where $T_{max}$ is the episode length. Each agent is located initially at the start vertex and is assigned to the current goal vertex. If it reaches the latter before the episode ends, it is immediately assigned another goal vertex. We assume that the *goal assignment* unit is external to the system, and the agents' behavior does not influence the goal assignments. An agent can reach the goal by performing the following actions: wait at the current vertex, and move to an adjacent vertex. The duration of each action is uniform, i.e., 1 time step. We assume that the outcomes of the actions are deterministic and no inaccuracies occur when executing the actions.

Each agent has complete knowledge of the graph $G$. However, it can observe the other agents only *locally*. When observing them, *no communication* is happening. Thus an agent does not know the (current) goals or intended paths of the other agents. It observes only their locations. The observation function can be defined differently depending on the type of graph. In our experiments, we use 4-connected grids and assume that an agent observes the other agents in the area of the size $m \times m$, centered at the agent's current position.

Our task is to construct an individual policy $\pi$ for each agent, i.e., the function that takes as input a graph (global information) and (a history of) observations (local information) and outputs a distribution over actions. Equipped with such policy, an agent at each time step samples an action from the distribution suggested by $\pi$ and executes it in the environment. This continues until time step $T_{max}$ is reached when the episode ends. Upon that, we compute the *throughput* as the ratio of the episode length to the number of goals achieved by all agents. This metric is used to compare different policies: we say that $\pi_1$ outperforms $\pi_2$ (in a particular episode) if the throughput of the former is higher.

**Partially Observable Markov Decision Process**   We consider a partially observable multi-agent Markov decision process [34, 35]: $M = \langle S, A, U, P, R, O, \gamma \rangle$. At each timestep, each agent $u \in U$, where $U = 1, \ldots, n$, chooses an action $a^u \in A$, forming a joint action $\mathbf{j} \in \mathbf{J} = J^n$. This joint action leads to a change in the environment according to the transition function $P(s'|s, \mathbf{j}) : S \times \mathbf{J} \times S \to [0, 1]$. After that each agent receives individual observations $o^u \in O$ based on the global observation function $G(s, a) : S \times A \to O$. And individual reward $R(s, u, \mathbf{j}) : S \times U \times \mathbf{J} \to \mathbb{R}$, based on the current state, agent, and joint action. To make decisions each agent maintains an action-observation history $\tau^u \in T = (O \times A)^*$, which is used to condition a stochastic policy $\pi^u(a^u|\tau^u) : T \times A \to [0, 1]$. The task of the learning process is to optimize the policy $\pi^u$ for individual each agent in order to maximize the expected cumulative reward over time.

## 4   Learn to Follow

The suggested approach to solve the considered LMAPF problem, which we dub FOLLOWER, comprises of the two complimentary modules combined into a coherent pipeline shown in Fig. 2.

---

[1]In MAPF literature, a plan is typically denoted with $\pi$. However, in RL, this is reserved to denote the policy. As we use both MAPF and RL approaches in this work, we denote a plan as $p$.

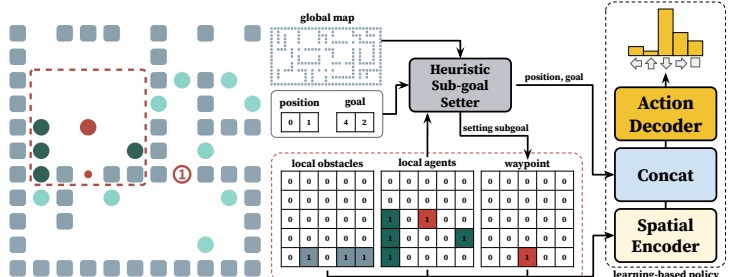

Figure 2: The general pipeline of the FOLLOWER approach. The action selection policy for each agent is decentralized and consists of two modules: Heuristic Sub-goal Decider, which address long-term path planning problem and Learning-based Policy optimization module, which addresses the short-term conflict resolution task.

First, a *Heuristic Sub-goal Decider* is used to construct an individual path to the goal and choose a waypoint on this path that becomes the agent's local goal, which we also call a sub-goal. Second, a *Learnable Follower* is invoked to reach the sub-goal. This module decides which actions to take at each time step until the sub-goal is reached or until the agent gets too far away from it. In both cases the sub-goal decider in called again and the cycle repeats.

## 4.1 Heuristic Sub-goal Decider

In essence the purpose of this module is to provide a waypoint (sub-goal) in the vicinity of the agent, pursuing which will allow agent to progress towards its (global) goal. A conventional heuristic search algorithm, i.e. A*, is used to construct a path to the latter from the current location. Global information on the locations of the static obstacles, i.e. the map, is used for pathfinding. The other agents are not taken into account at this stage, thus the constructed path may go through them. Once the path is built a node node located $K$ steps away from the current position is chosen as the current sub-goal. Here $K$ is the user-specified parameter.

An crucial design choice for this module is what individual path to build. On the one hand, A* finds the shortest (individual) path to the goal. On the other, as we noted empirically, when the number of agents is very high and each agent is following the shortest path, congestion often arise in the bottleneck parts of the map, such as corridors or doors. This degrades the performance dramatically. To this end we suggest to search not for the shortest paths but rather the evenly dispersed paths. This is implemented as follows.

At each time step the information on the locations of the locally observed agents is stored, in what we call a heatmap, and used to compute the additional transition costs for individual pathfinding. The number of times the other agents were seen in a certain location (grid cell in our experiments) is multiplied by the user-defined parameter $C$ and added to the transition cost to that location. Intuitively, if many agents are noticed in particular areas of the map the transition costs of the latter are increased so A* will avoid them. This balances the distribution of the agents' paths across the map and contributes to collision avoidance. Indeed, each agent maintains its own heatmap and performs pathfinding individually, thus the assumption that the agents do not share any data is not violated.

## 4.2 Learnable Follower

This module implements a learnable policy that is tailored to achieve the provided sub-goals while avoiding collision with the other agents. The policy function is approximated by a (deep) neural network and, as the agents are assumed to be homogeneous, a single network is utilized during training (a technique referred to as *policy sharing*). This approach is beneficial for complex tasks and large maps where it would be infeasible to learn a separate neural network for each agent, as the number of parameters increases linearly with the number of agents.

The input to the neural network represents the local observation of an agent and is comprised of a $3 \times m \times m$ tensor, where $m \times m$ is the observation range. The channels of the tensor encode the locations of the static obstacles, other agents and the current sub-goal respectively – see Fig. 2. If the latter is out of the agent's field of view, it is projected into the nearest cell (similarily to [19]).

The input goes through the *Spatial Encoder* first, then *Concat block* combines both spatial and non-spatial features (the position of the agent on a map and its global goal). This is followed by an *Action Decoder* that uses the function $f$ for approximating the state using observation history (the positions of other agents and the presence of obstacles) to make a decision. The network's output is a probability distribution over possible actions.

The whole pipeline is trained with a policy optimization algorithm using the reward function separated into the two components: upon reaching a sub-goal an agent receives a small intrinsic positive reward of $r_s$, whose value was determined empirically; upon reaching the global goal a conventional RL reward $r_g = 1$ is received. If while reaching the current goal the agent goes too far away from it, the heuristic sub-goal decider is invoked again. This mechanism is helpful in scenarios when to progress towards the global goal it is actually more beneficial to make a detour to avoid congestion with the other agents. Practically wise, a goal is recalculated if the agent's distance from its target exceeds a certain threshold, which is determined by a hyperparameter $H$.

The task of the learning process is to optimize the shared policy $\pi_\theta^u$ (i.e. the same policy for each agent) to maximize the expected cumulative reward. During the training process, rollouts (sequences of observation and action pairs) are gathered asynchronously from multiple environments with varying numbers of agents. The shared policy $\pi_\theta$ (actor network) is continually updated using the PPO clipped loss [36]: $\max_\theta \frac{1}{N} \sum_{u=1}^n \sum_{\mathbf{j}} \sum_{\tau^u} \pi_\theta(a^u|\tau^u) \hat{A}_{\text{clip}}(\tau^u, a^u) - \beta H(\pi_\theta(\cdot|\tau^u))$.

Here, $\beta$ is a coefficient that controls the entropy $H$, and $\hat{A}$ denotes the unclipped advantage function calculated using returns $\hat{R}$ for each step $t$ with observation history $\tau^u$: $\hat{A}(\tau^u, a^u) = \hat{R}_t^u - V\phi(\tau^u)$, with $\hat{R}_t^u = \sum_{k=0}^{T-1} \gamma^k r_{t+k}^u$. Here, we have a shared critic value function $V_\phi$, which is optimized using the following equation: $\min_\phi \frac{1}{N} \sum_{u=1}^n \sum_{\mathbf{j}} \sum_{\tau^u} \left( V_\phi(\tau^u) - \hat{R}_t^u \right)^2$.

In practice, the observation history $\tau^u$ is effectively modeled using a recurrent neural network (RNN) integrated into the actor and critic networks. The actor network is parameterized by $\theta$, while the critic network is parameterized by $\phi$. In our approach, we specifically utilize the GRU architecture [37].

The introduced intrinsic reward function allows the efficient training of an agent using relatively short rollouts, as evidenced by our experimental results, which demonstrate that a rollout length of 8 is sufficient for training. This is crucial for ensuring lifelong learning, as episodes may not have a clear ending point.

During the inference phase, each agent uses a copy of the trained weights, and other parameters remain unchanged. The proposed FOLLOWER scheme, despite its simplicity, allows the agent to separate the two components of the overall policy transparently and does not require the involvement of any expert data for training. The learning process is end-to-end and the number of hyperparameters (such as $K$ and $H$) that affect the result is relatively small. Finally, the reward function used is simple and does not require involved manual shaping.

# 5 Experimental Evaluation

To evaluate the efficiency of the proposed method[2], we have conducted a set of experiments, comparing it with the existing learnable and search-based algorithms on different grid maps. The episode length was set to 512 in all experiments. The agents field-of-view was $11 \times 11$. When training FOLLOWER we used the following values of the reward components: $r_g = +1$ and $r_s = +0.1$. The *Spatial Encoder* was realized as ResNet neural model [38], the *Concat block* – as a Multi-Layer Perceptron (MLP), and the *Action Decoder* – as a recurrent neural network, separated for actor and critic and based on GRU [37]. In the experiments, values of 2 and 10 were used for $K$ and $H$ respectively. The weighting coefficient $C$ for sub-goal setter was set to $0.4$. More information about which (hyper) parameters were tuned and how is provided in the Appendix. After fixing all the parameters, the final policy was trained using a single TITAN RTX GPU in approximately 1 hour.

---

[2] We are committed to open-source FOLLOWER.

## 5.1 Comparison With the Learnable Methods

In the first series of experiments, we have compared FOLLOWER with the two state-of-the-art learnable MAPF solvers, i.e. PRIMAL2 [19] and PICO [22]. Similarly to FOLLOWER, both are decentralized and rely on the local observations of the other agents. However, PRIMAL2 assumes that the local observations contain not only information about the current locations of the agents but also about their goals on the global map. PICO assumes that the agents can communicate, through selected central agent. Recall that our solver has access neither to any information about the other agents except their current locations nor to communication between the agents.

As learnable methods assume training on a certain maps topology, we use the maps suggested by the authors of the respective baselines for a fair comparison. Specifically, we compare with PRIMAL2 on the maze-like maps of size $65 \times 65$ on which PRIMAL2 was trained, and we compare with PICO on the maps with random obstacles described in the PICO paper. The visualizations of the maps are given in Appendix. We used the readily available weights for PRIMAL2 neural network (from the authors' repository). PICO was trained by us using the open-source code of its authors. Our method, FOLLOWER, was trained using the hyperparameters described in Appendix, and only PRIMAL2 maps were used for training. When training FOLLOWER, we vary the number of agents in range: 16, 32, 64, 128. After the training phase, we run the solvers on ten different maze-like/random maps that were not used while training. Each map was populated with varying numbers of agents: from 2 to 256. The goals for LMAPF were generated and assigned to agents randomly.

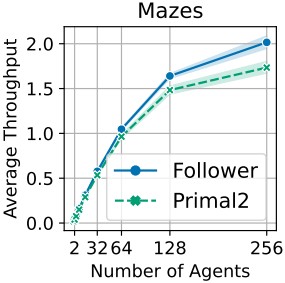

Figure 3: Average throughput on maze-like environments. The shaded area indicates 95% confidence intervals.

**FOLLOWER vs. PRIMAL2 results** are depicted on Fig. 3. The OX axis shows the number of agents and OY axis shows the average throughput. Indeed, when the number of agents is low both algorithms demonstrate similar results. However, with an increasing number of agents the gap in performance is getting pronounced. The throughput of the FOLLOWER is 11% better for 128 agents and 17% better for 256 agents. Overall, one can claim that despite having access to less MAPF-related data FOLLOWER outperforms PRIMAL2 when the number of agents is not low, i.e. in cases where the potential conflicts between the agents are not rare.

**FOLLOWER vs. PICO results** are presented in Table 1. Clearly, FOLLOWER demonstrates a superior performance across all scenarios. The poor performance of PICO can be attributed to the inherent difficulties in learning effective communication strategies for prioritizing large number of agents. Authors of PICO trained their method on 8 agents. We hypothesize that this limited population size may have impeded the acquisition of knowledge necessary for effective coordination among a larger number of agents. Both FOLLOWER and PRIMAL2 outperform PICO showing their ability to generalize (as they were not trained on PICO type of maps), with FOLLOWER being the ultimate winner.

Table 1: The comparison of FOLLOWER with PICO on random maps with different obstacle densities, taken from PICO evaluation setup.

| Algorithm | Agents | Obstacle Density | | | |
|---|---|---|---|---|---|
| | | 0% | 10% | 20% | 30% |
| FOLLOWER | 8 | **0.61** (±0.01) | **0.57** (±0.02) | **0.49** (±0.04) | **0.38** (±0.21) |
| PRIMAL2 | 8 | 0.44 (±0.03) | 0.39 (±0.04) | 0.3 (±0.05) | 0.19 (±0.11) |
| PICO | 8 | 0.19 (±0.01) | 0.18 (±0.03) | 0.14 (±0.04) | 0.05 (±0.05) |
| FOLLOWER | 16 | **1.1** (±0.03) | **0.96** (±0.05) | **0.85** (±0.19) | **0.56** (±0.34) |
| PRIMAL2 | 16 | 0.79 (±0.03) | 0.67 (±0.06) | 0.51 (±0.08) | 0.31 (±0.14) |
| PICO | 16 | 0.31 (±0.03) | 0.25 (±0.04) | 0.23 (±0.06) | 0.08 (±0.06) |
| FOLLOWER | 32 | **1.81** (±0.05) | **1.45** (±0.15) | **1.21** (±0.27) | **0.84** (±0.39) |
| PRIMAL2 | 32 | 1.25 (±0.04) | 1.02 (±0.11) | 0.71 (±0.12) | 0.4 (±0.16) |
| PICO | 32 | 0.46 (±0.05) | 0.35 (±0.1) | 0.28 (±0.12) | 0.12 (±0.09) |
| FOLLOWER | 64 | **2.6** (±0.11) | **1.88** (±0.33) | **1.24** (±0.23) | **0.71** (±0.36) |
| PRIMAL2 | 64 | 1.63 (±0.05) | 1.15 (±0.15) | 0.73 (±0.13) | 0.44 (±0.18) |
| PICO | 64 | 0.42 (±0.07) | 0.41 (±0.13) | 0.28 (±0.12) | 0.11 (±0.1) |

**Out of the distribution evaluation.**
An important attribute of any learnable algorithm is the so-called *generalization*, i.e. the ability to solve problem instances that are not alike to the ones that were used for training. We have already seen that FOLLOWER generalizes better than PRIMAL2 to the PICO type of maps (random ones). Now we run an additional test when we evaluated both algorithms on two (unseen while learning) maps from the well-known in the MAPF community

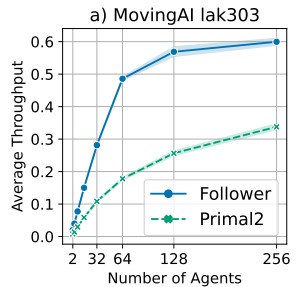 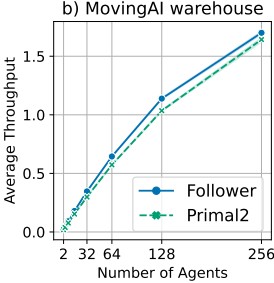

Figure 4: The results on a) `lak303d` and b) `warehouse` maps. The shaded area indicates 95% confidence intervals.

MovingAI benchmark [1]: `warehouse-10-20-10-2-1` and `lak303d`. The former map is $63 \times 171$ in size and represents the warehouse environment. It is similar to a certain extent to the maze-maps on which FOLLOWER and PRIMAL2 were trained. The latter map is a video-game map that was downscaled by us to have size $95 \times 95$. Its topology is quite different from the one of the maps used for training FOLLOWER and PRIMAL2. The results of these experiments are presented in Fig. 4. Note that we did not evaluate PICO on out-of-the-distribution maps due to its poor performance in the previous experiment.

On `warehouse-10-20-10-2-1` FOLLOWER and PRIMAL2 demonstrate similar performance, with our method achieving slightly higher throughput. I.e. the FOLLOWER'S throughput is 9.5% higher for 128 agents and 4% higher for the 256 agents. The results on `lak303d` are quite different. First of all, both algorithms have much lower average throughput compared to maze-like and warehouse environments. This is expected, as the topology of the game map differs a lot from the latter maps. Second, the throughput of FOLLOWER is significantly higher, providing another evidence (in addition to the results on PICO maps) that the generalization ability of our approach is better.

## 5.2 Comparison With the Centralized Search-based Solver

While most of the learnable approaches compare their results only with other learnable methods, we have also compared FOLLOWER with the state-of-the-art search-based algorithm for solving LMAPF – RHCR[3] [12]. In contrast to the proposed method, RHCR is a centralized approach that coordinates all the agents and does not restrict the observation and/or communication abilities of the agents. This planner has several parameters that influence its performance. We have varied the planning horizon $(2, 5, 10, 20)$, the re-planning rate $(1, 5)$ and found that the best results are achieved when the first parameter is set to 20 and the second one to 5. The time limit for re-planning was set to either 1 or 10 seconds. The MAPF solver used in RHCR was set to PBS [39]. The rest parameters were left default.

The comparison was conducted on the same `warehouse` map as in the original paper [12]. The possible placements of start and goal locations were also restricted in the same way as in the original paper. The number of agents in this experiment reached 192 as no more agents are able to be placed with the given restrictions to start locations. We generated 10 random instances per each number of agents.

Besides RHCR we have also evaluated different versions of our solver. First, we want to assess the impact of the learnable component on the FOLLOWER'S performance. To this end, we removed it from FOLLOWER and let each agent simply plan its path with A* and perform the first action. In case the path can not be found the action is selected randomly. We refer to this approach as Randomized A*. We evaluated two versions of Randomized A*: the one that treats the other agents (within field-of-view) as obstacles and the one that does not. Next, we were interested in how weighting the transition costs for A* affects the FOLLOWER'S performance. Thus, we have created a version of FOLLOWER that doesn't include this weighting (i.e. $C = 0$) and all transitions have uniform cost.

The results of this experiment are depicted in Fig. 5. Clearly both version of Randomized A* are outperformed by FOLLOWER. This confirms that the learnable policy is crucial to FOLLOWER. The introduced technique of penalizing the transitions to the areas where the other agents are often observed is also important, as in its absence the results of FOLLOWER are on par with Randomized A*.

---

[3]We used an implementation of RHCR from the authors' repository

Only combining the learnable module with the weighting technique we end with the best-performing method.

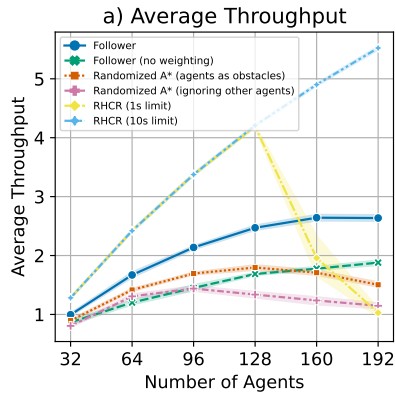

Figure 5: Average throughput on `warehouse` map. The shaded area indicates 95% confidence intervals.

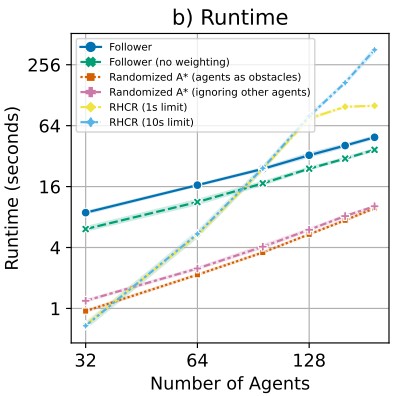

Figure 6: Average runtime on `warehouse` map.

Both versions of RHCR significantly outperform competitors in instances with up to 128 agents. However, when the number of agents increases to 160 and 192 the performance of RHCR with 1s time cap for re-planning degrades dramatically and it gets outperformed by FOLLOWER. This pinpoints the principal limitation of the centralized approach – it does not scale well to large number of agents when the time limit for finding a MAPF solution is imposed. To better understand how the runtime of the evaluated methods is affected by the increasing number of agents, see Fig. 6. Here each data point says how much time on average was spent to solve a LMAPF instance (on a single CPU, 1 thread). Indeed, FOLLOWER scales much better compared to RHCR. Moreover in practice it can be parallelized, i.e. run on each agent individually, while RHCR – can not.

## 5.3 Summary

The observed results let us infer the following conclusions. First, the suggested approach outperforms the learnable decentralized competitors, when it comes to a large number of agents and/or maps that are different from the ones used for training. Second, the learnable component of FOLLOWER is crucial to its high performance (as well as the introduced weighting technique). Third, when the number of agents is significantly high, FOLLOWER can outperform the centralized LMAPF solver when a (reasonable) time cap for the latter is introduced.

## 6 Conclusion

This study addresses the challenging problem of decentralized lifelong multi-agent pathfinding. The proposed FOLLOWER approach utilizes a combination of a planning algorithm for constructing a long-term plan and reinforcement learning for reaching short-term sub-goals and resolving local conflicts. The proposed method consistently outperforms decentralized learnable competitors in challenging scenarios. Moreover, our approach can show better results, compared to state-of-the-art centralized planner in certain setups. Directions for future research may include: enriching the action space of the agents, handling uncertain observations and external (stochastic) events.

## 7 Limitations

As in many other works on that topic (including the ones we compare with) we rely on the following assumptions. The map of the environment is accurate and the configuration of the static obstacles does not change. The agents are assumed to have perfect localization and mapping abilities. The agents execute actions accurately and their moves are synchronized. All these may be considered as the limitations as in real world, e.g. in robotic applications, many of the assumptions do not hold.

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
