## A Anonymized Code

The anonymized code of FOLLOWER is available at Learn-to-Follow-13223.

This repository contains everything needed to reproduce the results of FOLLOWER (train and eval scripts, pre-trained weights, dataset of maps, etc.).

Further in this Appendix we will refer to specific maps from our dataset by the names used in the repo.

## B Hyperparameters

Table 1 presents the hyperparameters of FOLLOWER. The hyperparameters for which the tuning range is given (e.g. learning rate, LSTM hidden size, $K$ (distance to the sub-goal), etc.) were optimized using Bayesian search. The observation radius was set to $11 \times 11$ as it is commonly used in similar learning-based methods (with whom we compare). The parameters for the number of *rollout workers*, *environments per worker*, and *training steps* were empirically determined to decrease the overall learning time of the algorithm. For the remaining paramaters (value loss coefficient, GAE$_\lambda$, activation function, network initialization) we used the default values provided in the SampleFactory framework[1].

We perform a hyperparameter sweep consisting of approximately 150 runs, totaling around 120 GPU hours. The final model was trained using a single TITAN RTX GPU in approximetely 1 hour.

Table 1: The hyperparameters of FOLLOWER. The *tune range* column indicates the range for parameters adjusted through a hyperparameter optimization procedure.

| Hyperparameter | Value | Tune range |
|---|---|---|
| Adam learning rate | 0.000123 | $0.0001 - 0.0002$ |
| $\gamma$ (discount factor) | 0.962983 | $0.95 - 0.99$ |
| Rollout | 8 | $[4, 8, 16, 32]$ |
| Clip ratio | 0.076785 | $0.05 - 0.2$ |
| Batch size | 1024 | $[1024, 2048, 4096]$ |
| Optimization epochs | 1 | $[1, 5, 10]$ |
| Entropy coefficient | 0.014733 | $0.01 - 0.02$ |
| Value loss coefficient | 0.5 | - |
| GAE$_\lambda$ | 0.95 | - |
| ResNet residual blocks | 4 | $[2, 4, 6, 8]$ |
| ResNet number of filters | 64 | $[32, 64, 128]$ |
| LSTM hidden size | 512 | $[128, 256, 512]$ |
| Activation function | ReLU | - |
| Network Initialization | orthogonal | - |
| Number of agents | $[16, 32, 64, 128]$ | - |
| Rollout workers | 8 | - |
| Environments per worker | 2 | - |
| Training steps | 60000000 | - |
| $K$ (sub-goal dist.) | 2 | $[1, 2, 3, 4, 5, 6]$ |
| $H$ (sub-goal recalc.) | 10 | $[2, 4, 8, 10, 12]$ |
| $C$ (add. transition cost) | 0.4 | $[0.0, 0.2, 0.4, 0.6, 0.8, 1.0]$ |
| Observation radius | $11 \times 11$ | - |

## C Weighted Transition Cost

As explained in the main body of the paper, each agent utilizes the heatmap of frequently used grid cells for individual pathfinding. The number of times the other agents were seen in a certain cell is multiplied by the user-defined parameter $C$ and added to the transition cost to that cell. This helps each agent to avoid areas that are often used by everyone and thus to pro-actively avoid congestion.

---

[1]github.com/alex-petrenko/sample-factory

The effect of incorporating additional transition cost is visualized in Fig. 1. Here the cells with the higher intensity of red indicate the areas visited (by the agents) more frequently during the episode. These heatmaps were constructed from solving a single instance with 128 agents with FOLLOWER as well as with Randomized A* (i.e. trimmed FOLLOWER lacking a learnable policy).

Clearly, when the additional transition costs are not employed, i.e. $C = 0.0$, the agents tend to use the central part of the map often. This makes it hard for the agents to avoid each other. On the other hand, when additional transition costs are applied, the agents get evenly distributed across the map, which prevents congestion and increase the performance (as confirmed by the experiments, reported in the main body of the paper).

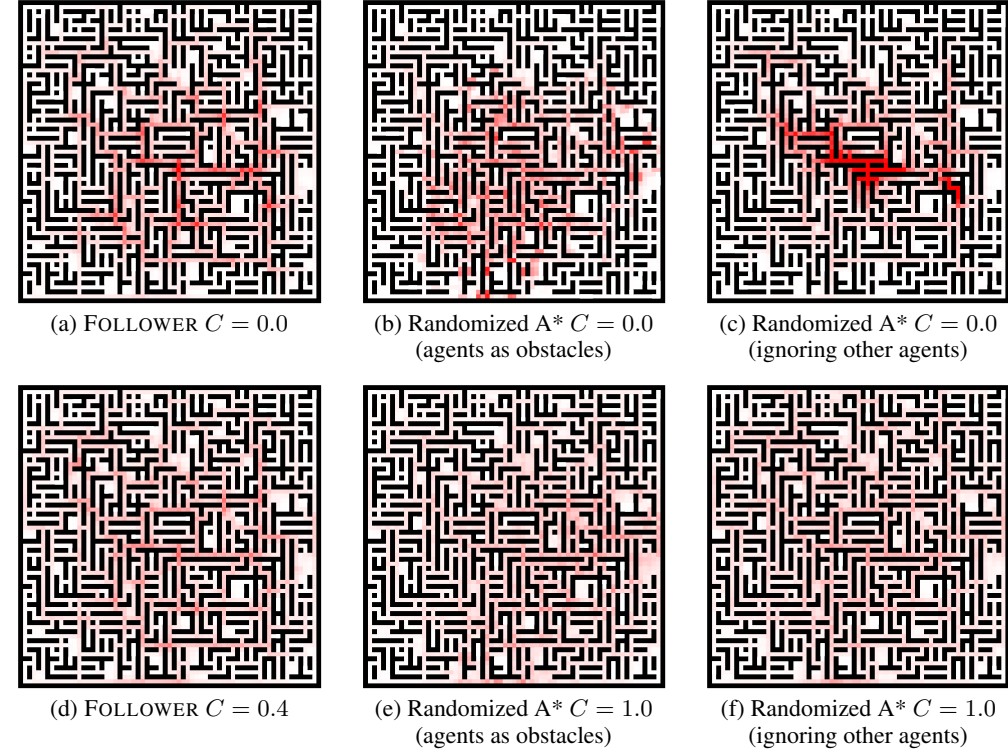

(a) FOLLOWER $C = 0.0$

(b) Randomized A* $C = 0.0$
(agents as obstacles)

(c) Randomized A* $C = 0.0$
(ignoring other agents)

(d) FOLLOWER $C = 0.4$

(e) Randomized A* $C = 1.0$
(agents as obstacles)

(f) Randomized A* $C = 1.0$
(ignoring other agents)

Figure 1: Heatmaps representations of how often the agents visited certain cells of the grid when solving a particular LMAPF instance containing 128 agents.

## D   Tuning RHCR Parameters

As it was mentioned in the main text, we have tuned the parameters of RHCR before conducting the empirical comparison to our method. We varied the values of such RHCR parameters as planning horizon (2, 5, 10, 20), re-planning rate (1,5) and time limit for each re-planning attempt (1s, 10s). Planning horizon parameter controls for how many time steps the resultant plans will be collision-free. E.g. when it equals 10 it is guaranteed that for the next 10 time steps the agents following the constructed plans will not collide. Re-planning rate determines how frequently (in time steps) reconstruction of the plans (for all agents) occurs. Time limit parameter restricts the amount of time (in seconds) which is alotted for each re-planning attempt.

Fig. 2 demonstrates the results of different versions of RHCR (note that planning horizon cannot be lower than re-planning rate). The best average throughput was achieved by RHCR with re-planning rate 5 and planning horizon 20 (denoted as ($w = 5$, $h = 20$) in the figure). The same values of these parameters were also used for the experimental evaluation of RHCR on the `warehouse` map in the original paper. Thus, the results of this version were included into the main part of our paper.

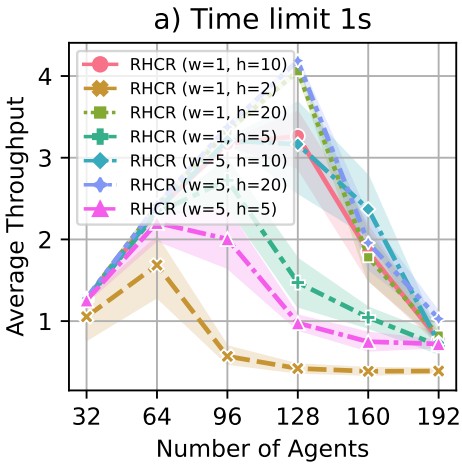
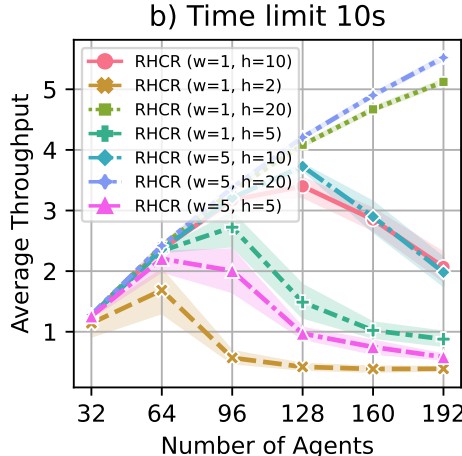

Figure 2: Evaluation of RHCR with varied parameter settings: $w$ – re-planning rate, $h$ – planning horizon.

## E    Impact Of the Episode Length

We set the episode length to 512 in all experiments. Additionally, we examined the impact of episode length on the throughput of FOLLOWER by running additional experiments on the maze maps with 256 agents. The results are shown in Fig. 3. The throughput increases first, starting from 1.43, but then plateaus at 1.65. We attribute the initial increase to the accumulation of knowledge regarding the transition cost. We believe our choice of the episode length (512) is reasonable.

## F    Maps Visualizations

Fig. 4 illustrates examples of the maps used for testing. The names of the maze-like maps and Pico-maps are the same as in the repository.

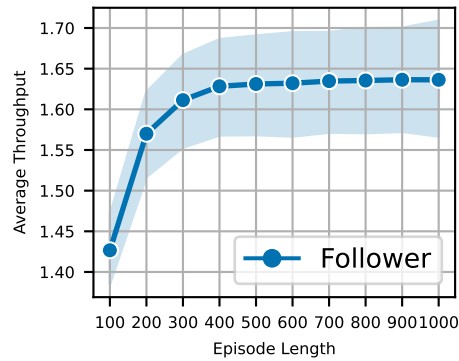

Figure 3: Impact of episode length on throughput of FOLLOWER with 256 agents.

Initial positions of the agents are represented by the filled circles, while their (initial) goals are represented by the empty ones. Each agent is assigned a unique goal initially. Subsequent LMAPF goals are randomly generated, ensuring a feasible path from the agent's current location to the goal exists. The goals for each agent are generated independently using a fixed seed, ensuring consistency and enabling fair testing of the algorithms (i.e. each algorithm gets the same start/goals locations).

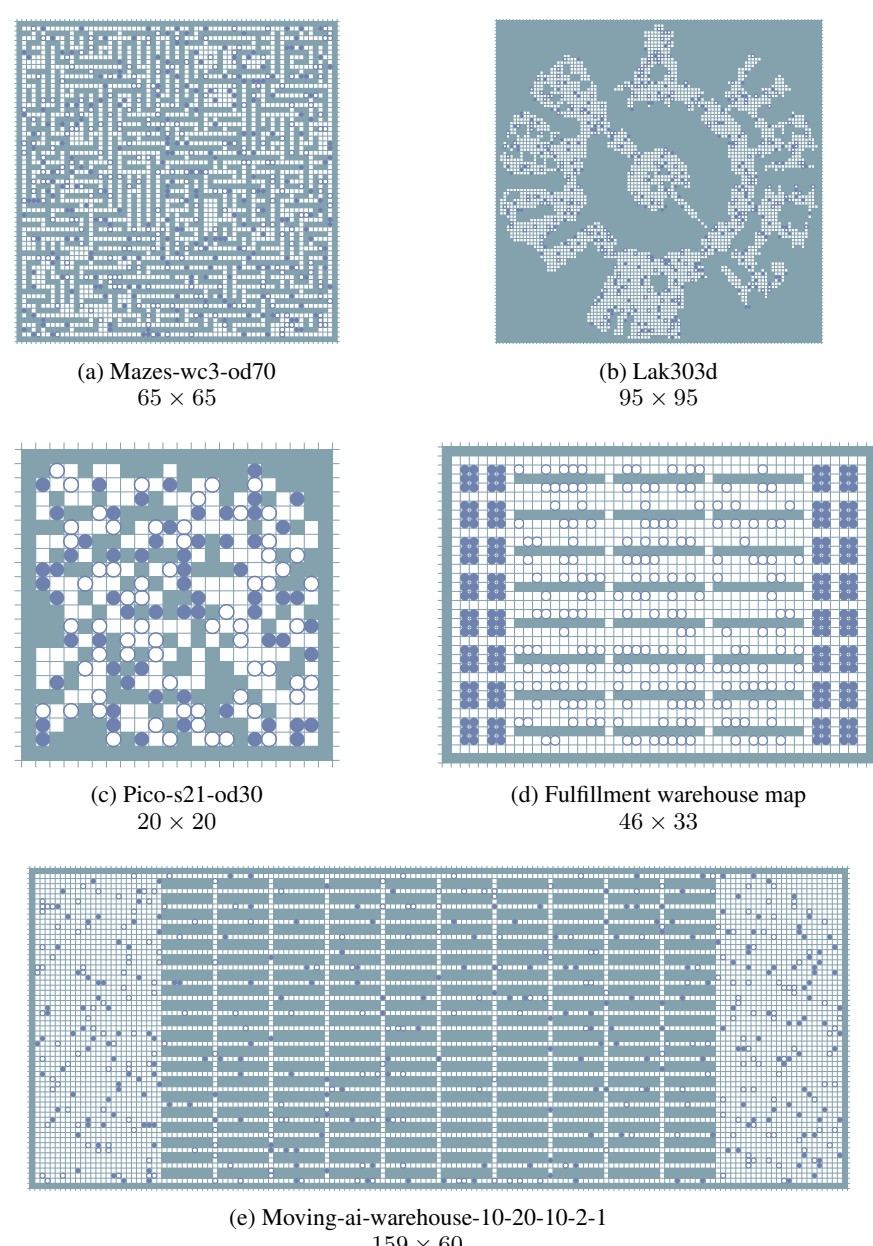

(a) Mazes-wc3-od70
$65 \times 65$

(b) Lak303d
$95 \times 95$

(c) Pico-s21-od30
$20 \times 20$

(d) Fulfillment warehouse map
$46 \times 33$

(e) Moving-ai-warehouse-10-20-10-2-1
$159 \times 60$

Figure 4: Visualizations of all the maps with a maximum number of agents used during the experimental evaluation.

## G Detailed Comparison with PRIMAL2

The detailed results of a comparison between FOLLOWER and PRIMAL2 on the entire test set of maze maps are illustrated in Fig. 5. FOLLOWER demonstrates superior performance on all maps. It is important to note that the PRIMAL2 algorithm utilizes various heuristics to take advantage of the topological characteristics of the maps (i.e. the presence of corridors).

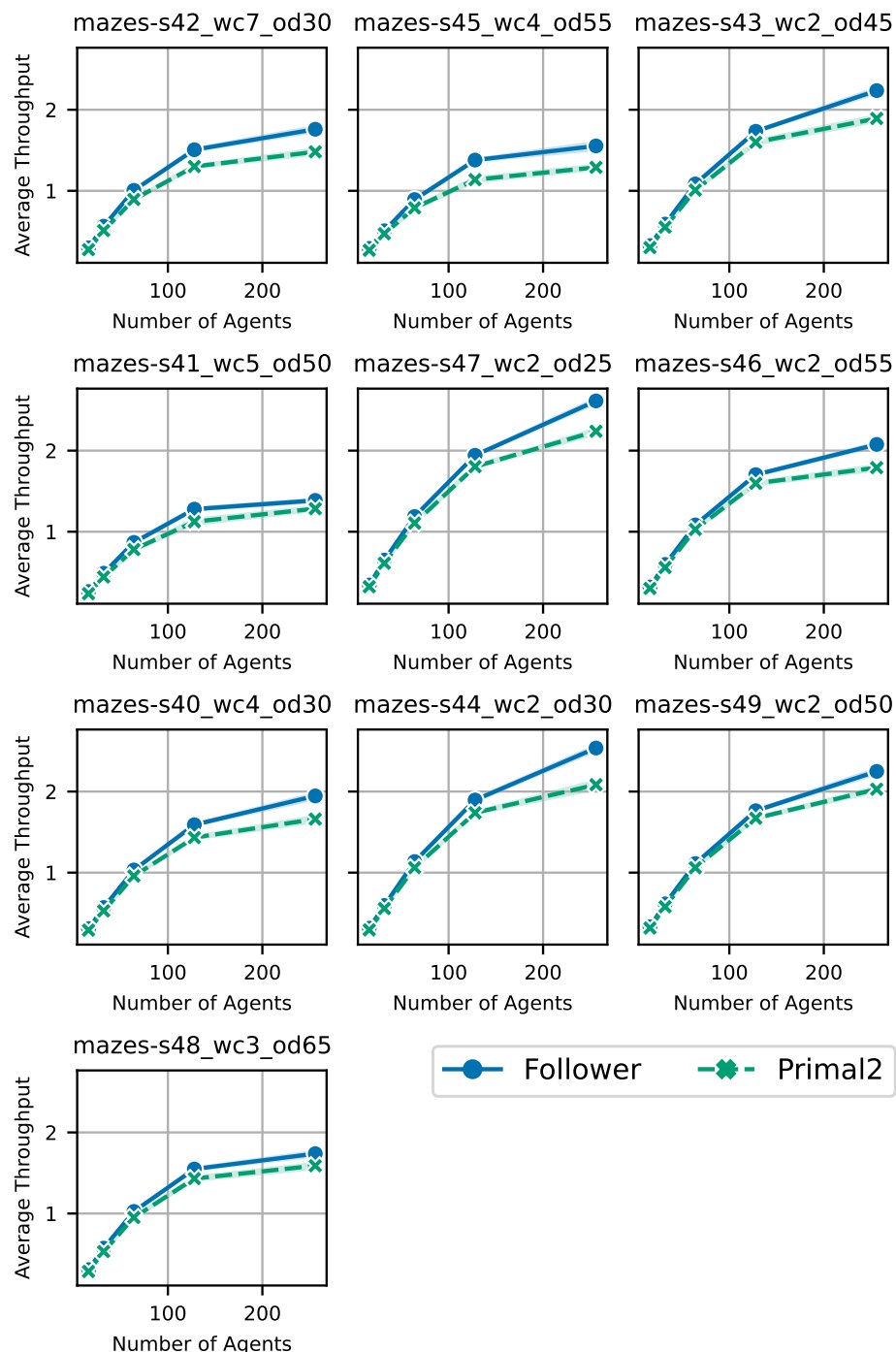

Figure 5: Average throughput on entire test set of the maze-like environments. The shaded area indicates 95% confidence intervals.