# OpenReview forum: "Learn to Follow: Lifelong Multi-agent Pathfinding with Decentralized Replanning"
_NeurIPS.cc/2023/Conference — Submitted to NeurIPS 2023_

### Official Review · Reviewer_VcwW · 2023-06-13

**Soundness:** 2 fair
**Presentation:** 3 good
**Contribution:** 2 fair
**Rating:** 4
**Confidence:** 3

**Summary:**

This paper addresses the multi-agent pathfinding problem by proposing an approach that utilizes a combination of a planning algorithm for constructing a long-term plan and reinforcement learning for reaching short-term sub-goals and resolving local conflicts. The results show that the proposed method outperforms decentralized learnable competitors and centralized planner.

**Strengths:**

1. The method is straightforward and concise.
2. The writing is clear and easy to understand.

**Weaknesses:**

1. The proposed method follows a hierarchical reinforcement learning framework, which has been extensively studied in previous works. There are limited contributions to the design of sub-goal selection.
2. In the heuristic sub-goal decider, A* is used to construct a path, which requires global information. As the sub-goal decider will be used multiple times during the episode, the overall method seems not fully decentralized.


**Questions:**

1. How the non-stationary problem of multi-agent reinforcement learning is addressed? As the policy directly optimizes the r^i.
2. Could the authors show the makespan of all methods?
3. As the policy is shared, could the authors explain how the agents handle situations where their paths cross? Given that the shared policy tends to result in similar actions, there is a possibility that the agents might end up both staying still or moving together into a collision.

**Limitations:**

This work has little negative societal impact.

---

> ### Author Rebuttal · Authors · 2023-08-09
>
> ### Weakness 1:
>
> Thank you for pointing out the familiarity of the hierarchical reinforcement learning framework in previous research. We acknowledge that the hierarchical RL framework has indeed been a subject of extensive investigation in the literature. However, in our work, we advance beyond the conventional application of hierarchical RL by introducing a distinctive emphasis on sub-goal selection within the context of multi-agent interactions in the MAPF domain.
>
> Our method introduces a departure from the typical hierarchical RL paradigm by allowing the Follower agent to pursue and accomplish multiple sub-goals along its trajectory. Unlike traditional approaches where an agent focuses on a single sub-goal per episode, our approach recognizes the potential benefits of considering a sequence of sub-goals. This opens up new avenues for more sophisticated decision-making policies, such as avoiding conflicts with other agents or strategically deferring immediate rewards for the sake of higher cumulative rewards.
>
> Furthermore, it's worth noting that the application of hierarchical RL to MARL has not been as extensively explored [1]. In our work, we successfully apply this technique to the challenging MAPF domain, offering a significant advancement over many existing works that primarily focus on simpler, toy examples. This demonstrates the practical relevance and effectiveness of our approach. Once again, we appreciate your comment as it certainly holds merit for inclusion in the methodological and related work sections.
>
> [1] Pateria, Shubham, et al. "Hierarchical reinforcement learning: A comprehensive survey." ACM Computing Surveys (CSUR) 54.5 (2021): 1-35.
>
> ### Weakness 2:
>
> Our method, Follower, is decentralized in a sense that it can be executed on a single agent and no communication with the other agents and/or central controller is needed (as Follower does not need to know the goals/paths/actions of the other agents). Indeed, Follower uses A* under the hood to find a path to the goal. We assumed in this work that each agent knows the static map of the environment and this map is utilised in A*. However, even if the full static map of the environment is not available we can easily substitute A* with one of its numerous variants tailored to partially-observable maps, like D*lite. This still will keep Follower decentralised (in the sense described above).
>
> ### Question 1:
>
> The issue of non-stationarity within multi-agent environments represents a crucial challenge in the realm of Multi-Agent Reinforcement Learning (MARL). In our work, we use the implementation of the PPO algorithm with a decentralized critic. In a number of papers [1,2], it has been shown theoretically and empirically that such an implementation in small-sized problems copes well with the problem of non-stationarity. Our work proposes integration with a high-level planner, which reduces a high-dimensional non-stationary task (with up to 128 agents) to a set of small-sized non-stationary tasks, which PPO with a decentralized critic successfully handles.
>
> [1] Yu, Chao, et al. "The surprising effectiveness of ppo in cooperative multi-agent games." Advances in Neural Information Processing Systems 35 (2022): 24611-24624.
>
> [2] Sun, Mingfei, et al. "Trust region bounds for decentralized ppo under non-stationarity." Proceedings of the 2023 International Conference on Autonomous Agents and Multiagent Systems. 2023.
>
> ### Question 2:
>
> Makespan is a performance indicator that is not directly applicable to lifelong MAPF. This measure is used in case of single-shot MAPF. To show the makespan we have modified the code of the algorithms to solve single-shot MAPF instances and run an additional experiment. The evaluation was made on the maps/instances taken from PICO’s repository with 20x20 grid size and 30% density of obstacles. Follower and PICO were not retrained for this type of instances, while for PRIMAL2 we took the weights provided by the authors, that were specifically trained for single-shot MAPF. The episode length was set to 256. In case if the algorithm was not able to find a solution within the given amount of steps, the makespan for the corresponding instance was set to 256. The obtained results (presented in the attached file of Author Rebuttal section) demonstrate that Follower significantly outperforms the competitors on the instances with up to 32 agents, while on the instances with 64 agents all the approaches demonstrate poor performance. Overall, out of all the instances that were solved by at least one of the evaluated approaches, Follower has found a better solution in ~83% cases, PICO - ~13%, PRIMAL2 - ~4%.
>
> ### Question 3:
>
> Though the policy is shared between the agents, it is conditioned on each agent’s current sub-goal. This sub-goal is determined using the agent’s individual global goal (which is unique and not shared between the agents) and the cost penalty heat map (which is also unique for each agent as it is constructed from the individual experience of an agent). In addition to this each agent has its unique observation history $\tau^u$, which aids him to prevent conflicts and make informed decisions based on its past experience. Finally, the policy is stochastic, meaning that each agent samples an action from action distribution, allowing the agents to make different actions in the same situations.
>
> These concepts are well demonstrated in the examples provided in our supplementary materials, specifically in Appendix A. The appendix contains a link to an anonymized repository (code in repository is unchanged since initial submission) that includes examples of animations showcasing the effectiveness of our approach in conflict resolution.

---

> > ### Comment · Reviewer_VcwW · 2023-08-17
> >
> > Thank you for the response. However, the contribution still seems more focused on applying hierarchical RL to a specific task like MAPF.

---

> > > ### Author Response · Authors · 2023-08-18
> > > **HRL and the MAPF task**
> > >
> > > We thank the reviewer for engaging in the discussion and sharing the post-rebuttal opinion. First, we note that most of the initially raised concerns (W2, Q1, Q3, Q4) seem to have been adequately addressed by us in the rebuttal, as they are not mentioned in the reviewer’s reply. Thus, we wish to discuss the remaining concern (W1) regarding the novelty of our approach in the context of HRL.
> > >
> > > First, we would like to note that we are not positioning our method as an HRL approach due to fundamental differences from classical HRL methods like the Options framework and the Feudal approach. While in HRL, one of the main ideas is to allocate sub-tasks specific to the environment (like ‘moving to the door’) and to learn low-level abstract policies (skills) that can be reused during further learning, a set of different skills, tailored to different sub-tasks, is not formed in our approach. In our approach, sub-goals reduce the sparsity of the reward function and allow a low-level RL-based policy to focus not on the pathfinding aspect but on conflict resolution (a very important skill that is hard to design in a deterministic/heuristic fashion). Still, we agree that our approach is relevant to HRL and can potentially give an impetus to developing new methods within HRL. Please also note that the number of works where (classical) HRL is applied to MARL is very limited. In most of them, toy environments with few agents are considered.
> > >
> > > Second, it is true that we have been focused on a specific multi-agent problem setting, i.e., multi-agent pathfinding (MAPF). The choice is not arbitrary but rather well-grounded. This setting is particularly challenging as practical scenarios may involve dozens, hundreds, and thousands of simultaneously acting (moving) agents. The desired output, i.e., policy, should be highly-generalizable (to unseen instances and types/topologies of maps) as in MAPF, we are interested not in solving decentralized POMDP in a certain environment (in RL terminology) but rather in an (apriori unknown) distribution of the environments (i.e., different maps). To our knowledge, no current MARL methods can efficiently solve this nontrivial problem.
> > > Moreover, MAPF is a very hot topic in the search community (as conventional ‘golden standard’ search techniques like A* struggle in multi-agent settings and specific involved machinery should be introduced to cope with the curse of dimensionality). However, the methods the researchers from this community typically develop are centralized and therefore do not scale well to many agents. We suggest tackling this problem by including the RL techniques in the loop. Of course, we are not the first to follow this line. However, as the paper shows, our approach leads to a policy that consistently outperforms state-of-the-art (in the learnable MAPF). In this context, we would like to mention also the well-known in the community Flatland competition [1] that has been held several times and was an official NeurIPS contest in 2020. This competition assumes solving a variant of the MAPF problem (similar to the one considered in our paper) without restricting methodology, i.e., both learnable and search-based solvers are allowed. Nevertheless, as the results of the previous competitions have shown,  the learnable methods (RL) seriously lagged behind the classical ones in their performance. Our work bridges the gap between these approaches and demonstrates how two approaches can leverage each other when combined thoughtfully.
> > >
> > >
> > > [1] https://arxiv.org/abs/2012.05893

---

### Official Review · Reviewer_wG3W · 2023-07-06

**Soundness:** 2 fair
**Presentation:** 3 good
**Contribution:** 3 good
**Rating:** 6
**Confidence:** 3

**Summary:**

This paper introduces a decentralized hierarchical approach without agent-to-agent communication for Lifelong Multi-agent Pathfinding (MAPF). The framework adds a congestion-based heuristic to an A*-planner and a low level Reinforcement Learning (RL) - based controller to follow the provided sub-goals. Experimental results show that the proposed method has a higher throughput (or rate of reach of new goals) for a range of maps.

**Strengths:**

- Easy to understand. Uses the well-studied A* planner with additional heuristics and a low level RL-based controller simply trained to reach goals promoting long term performance.
- Hierarchical framework is simple and could be effective way for decentralized control in an MAPF problem.

**Weaknesses:**

- The change of the heuristic in the A* planner seems weakly substantiated. While empirical results are promising, the need for hyperparameter tuning for the score and lack of guarantees on behavior may impede the use of this new heuristic.
- Confidence intervals for higher density experiments may be too large to claim better performance. (E.g. Table 1, 16 agents, proposed approach has throughput $ 0.56 \pm 0.34$ vs Primal2 having $0.31 \pm 0.14$ ). This may point to noisier behavior in the presence of more obstacles.

**Questions:**

1. Does the learnable follower explicitly handle collisions between agents? If I understand correctly, it is rewarded for reaching the goal (sub-goal and global) thus implicitly handling collision-avoidance.
2. What other metrics of interest are there apart from throughput? Is there a measure of the success rate of the given algorithm on a map such as mentioned in Primal2 [1] ?
3. Some typos/clarifications?
    1. L205 : “An crucial” → “A crucial”
    2. L207: “congestion often arise” → “congestion often arises”
    3. L238: “If while reaching the current goal the agent goes too far away from it,” → This is referencing the global goal?
    4. L263: “the reward function used is simple
    and does not require involved manual shaping.” → L237 mentions an empirically determined reward so this appears incorrectly stated.

References:

[1] PRIMAL2: Pathfinding via Reinforcement and Imitation Multi-Agent Learning - Lifelong, Damani et al, RAL 2020

**Limitations:**

- Access to global map is assumed for use of the A* algorithm.
- Several empirically determined reward function portions may hinder generalizability to different maps.

---

> ### Author Rebuttal · Authors · 2023-08-09
>
> ### Weakness 1:
>
> On the one hand, the suggested penalizing-transitions technique for A* does not violate any of its properties (i.e. A* with such a modified cost function is still guaranteed to be complete and to find the cost-optimal path). On the other hand we agree that this technique has no strict theoretical guarantees to increase the performance of Follwoer w.r.t. throughput. Still it provided a substantial boost in performance of Follower in all the scenarios that were evaluated. Moreover, there was no tuning of the penalty transition hyperparamter (i.e. C) for each of the experiments on different maps. Instead, this hyperparameter (as well as all the others) was tuned only once - during training on maze-like maps.
>
> ### Weakness 2:
>
> You're correct; agent behavior and results become more erratic as the number of obstacles increases. Additionally, significant variations arise from the fact that the outcomes are averaged across different maps with identical density. Despite the consistent maps density, their complexities differ considerably.
>
> To address this issue, we have conducted an additional experiment on randomly generated maps with a density of 30%. We increased the number of different LMAPF instances to 100 per map per number of agents, in contrast to the 10 instances mentioned in the submitted version of the manuscript. Comprehensive box-and-whisker plots are presented in Figure 2, which is enclosed in the attached document. Overall, the results clearly show the superiority of Follower.
>
> ### Question 1:
>
> Yes, you are right. The agent handles collisions implicitly through sub-goals and a global goal reward signal. The collision of agents results in them staying in their positions and not receiving rewards. The absence of this positive signal can be interpreted as an implicit method of collision avoidance.
>
> ### Question 2:
>
> In the original Primal2 the authors considered two MAPF scenarios: single-shot and lifelong. In the single-shot scenario when an agent reaches the goal it never receives a new one. In this case it is possible (and natural) to consider such measures as Success Rate and Makespan. In the lifelong variant of MAPF each agent upon arriving to the goal cell is immediately assigned a new one, thus the measures from the single-shot MAPF can not be straightforwardly applied. The most common measure of success for Lifelong MAPF is the Throughput. It is this measure that we used in our evaluation as in our work we are focused on the Lifelong MAPF. Please note that in the Primal2 paper the authors also use only the Throughput for measuring the success of the lifelong MAPF (and use the other measures for the single-shot MAPF only). Still, for this rebuttal we have performed an additional experiment comparing Follower, PICO and Primal2 on single-shot instances on random maps with 30% density of obstacles. Follower and PICO were not retrained for this type of instances, while for Primal2 we took the weights provided by the authors, that were specifically trained for single-shot MAPF. The plots regarding the obtained success rate and makespan are presented in the attached file (Author Rebuttal section). In brief, Follower outperformed other approaches in this experiment.
>
> ### Question 3:
>
> (1, 2, 4): Thank you, we will fix these shortcomings.
>
> (3): *"If while reaching the current goal the agent goes too far away from it"*
> Is referenced to sub-goal. We will explicitly state it in the main text.

---

> > ### Comment · Reviewer_wG3W · 2023-08-15
> > **Thanks for the response**
> >
> > I appreciate the response provided the authors and the additional experiments providing evidence that even with considering noise, their approach performs reasonably better than other MAPF algorithms like Primal2. The point that the hyperparameters tuned on one set of maps worked on different maps during test time is also impressive. For this reason I am willing to raise my score to a weak accept.

---

> > > ### Author Response · Authors · 2023-08-18
> > >
> > > We thank the reviewer for their involvement in the discussion. We are glad our rebuttal cleared the points raised in the initial review. We are committed to improving the paper following the reviewer’s comments.

---

### Official Review · Reviewer_prAH · 2023-07-07

**Soundness:** 3 good
**Presentation:** 3 good
**Contribution:** 3 good
**Rating:** 6
**Confidence:** 3

**Summary:**

The paper considers a decentralized multi-agent pathfinding (MAPF) problem. The main idea is to combine heuristic-based search and reinforcement learning. This work first determines subgoals and uses this information as intrinsic rewards. Empirically, it outperforms two baselines in the literature,  PRIMAL2 and PICO, in domains with different sizes and different numbers of agents. The method also demonstrates the ability to generalize to domains unseen during training.

**Strengths:**

The main observation is that pure heuristic search would not have a good performance in complex domains, where collaborative behaviors like congestion avoidance may not emerge.
This work shows an inspiring combination of heuristic-based search and reinforcement learning.

The proposed algorithm also outperforms a centralized control algorithm (RHCR) when the number of agents is large or when the computational budget is small (so the centralized algorithm is expensive).


**Weaknesses:**

Some concerns about the practicality of the algorithm: 1) It requires some hyperparameter selection. 2) It requires pre-train a neural model. When it outperforms RHCR in some settings when RHCR is run for 10s, we need to consider the computational overhead of running the RL algorithm during training.
In terms of performance, the performance between FOLLOWER and PRIMAL2 is very close in some domains.

**Minor points.** Line 203, duplicate “node.”


**Questions:**

Are the concerns in the weakness section correct? In case I misunderstood the results.

Usually hand-design the Intrinsic reward is difficult. It’s possible that the agent keeps collecting intrinsic rewards without reaching the real goal. Have the authors tried different intrinsic reward values?

It seems that both PRIMAL2 and PICO have access to more information than FOLLOWER (information about goals on the global map, communication between agents), but FOLLOWER still outperforms the baselines?

**Limitations:**

The authors have mentioned the limitations of this work to be the assumptions of static environment, perfect perception.

---

> ### Author Rebuttal · Authors · 2023-08-09
>
> ### Weakness 1:
>
> Hyperparameter selection is a necessary part of almost any learnable method. Moreover, in the path planning domains even the non-learnable search-based methods often require setting their parameters. E.g. the state-of-the-art search-based LMAPF solver with whom we compare, i.e. RHCR, requires setting the re-planning frequency, planning horizon etc. (It is worth mentioning here that we did try different values of the RHCR parameters and finally picked the ones that performed the best). Meanwhile, we tuned the hyperparameters of Follower to optimize the throughput on the maps from the training dataset, i.e., only maze maps from Primal2 paper. Then, at the test time we did not re-tune any hyperparameter, they all were left the same as when training had been finished. Subsequently, the results of Follower on out-of-distribution maps (i.e., on maps with topologies that it did not see during training) clearly demonstrate that Follower can generalize to unseen problem instances without requiring parameter re-tuning.
>
> ### Weakness 2:
>
> Regarding the computational overhead, we acknowledge that training a neural model requires resources, and running the RL algorithm during training can add to the computational burden. However, we want to clarify that the Follower model was trained only once using the training set of the maze maps. The evaluation phase involved using the same pre-trained model with frozen weights for all experiments, including experiments on the maps that notably differ in their topology from the ones used for training. There was no additional training performed on these maps during evaluation. Therefore, we believe it is correct not to consider the training time of the Follower as a computational overhead during the evaluation phase.
>
> ### Weakness 3:
>
> In terms of performance, Follower and Primal2 exhibit very close results in the maze-like domains, such as mazes and warehouse maps. It is worth noting here that Primal2 was tailored specifically to reason about corridors and possible deadlocks occurring in them (e.g. see Figure 3 of the Primal2 paper), while our method is not (as we purposefully avoided any narrow specification of our approach). That means that even without special corridor reasoning Follower is able to be on par with Primal2 in corridor-rich domains and even outperform it. When the environment is not maze-like, e.g. on random maps, Follower’s superiority is clearly visible. Moreover, for this rebuttal we additionally conducted extra experiments on 6 maps of varying topology (without any re-training of Follower/Primal2) and, again, Follower significantly outperformed Primal2 on each of these maps - see Fig.1 in the attached file (see Author Rebuttal section) for the detailed plots .
>
> ### Question 1:
>
>
> Yes, designing the reward function is a tricky process. Thus, in our research, we tried to keep it as simple as possible. We used only two reward components: $r_g = 1$ and $r_s = 0.1$, for reaching the goal and subgoal, respectively. We believe these rewards are easy to interpret. Moreover, we did not specifically tune these values in the hyperparameter search procedure; instead, we used the values determined during preliminary experiments.
>
> ### Question 2:
>
> Yes, formally, Follower has access to less information than Primal2 and PICO. However, in our experiments, it outperforms the latter. We believe this is due to a proper combination of both search-based and learning-based components, as suggested by us in the paper: using search to find a path and utilizing an RL-policy to navigate along this path. Both Primal2 and PICO also incorporate demonstrations during training, which could introduce additional inductive bias to their policies.

---

> > ### Comment · Reviewer_prAH · 2023-08-16
> > **Thanks for the response**
> >
> > Thanks to the authors for the responses and the additional results. In the additional experiments, it's clear that Follower outperforms the baseline algorithm. My questions have been answered.
> >
> > I believe this work has its merits. However, I think the authors also acknowledge in the general response that this HRL framework is not completely novel. It's also tailored for a very specific setting (lifelong MAPF). So I'll keep the score the same.

---

> > > ### Author Response · Authors · 2023-08-18
> > >
> > > We thank the reviewer for the discussion and are glad that our initial response (as far as we can get from the reply) has helped to answer all of the questions/concerns, i.e., W1, W2, W3, Q1, Q2, explicitly formulated in the original review. We have also elaborated on our motivation to focus on the MAPF problem and the novelty of the approach in reply to VcwW.

---

### Official Review · Reviewer_8Qnn · 2023-07-10

**Soundness:** 3 good
**Presentation:** 3 good
**Contribution:** 3 good
**Rating:** 5
**Confidence:** 3

**Summary:**

This paper proposes a novel method for decentralized lifelong MAPF. The method consists of two components: a heuristic sub-goal decider, which assigns sub-goals for each agent using a heuristic (e.g., A*), and a learning-based policy network, which outputs actions for achieving the short-term subgoals. The paper compares the proposed method with both learning-based decentralized methods and the search-based centralized method on extensive setups and demonstrates the proposed method's superiority. The paper also provides insightful ablation studies.

**Strengths:**

1. The idea of using heuristics to solve long-term planning and utilizing learning-based policies for achieving short-term sub-goal is reasonable and also commonly used in many other tasks.

2. The paper compares the proposed method with both learning-based decentralized methods and a search-based centralized method on extensive setups and demonstrates the proposed method's superiority and generalization capability.

3. The paper also provides insightful ablation studies and verifies the necessity of each proposed component.

4. The paper is well-written and easy to follow.

**Weaknesses:**

1. Since I'm not active in the MAPF field right now, I am unsure if there is literature sharing similar ideas in the MAPF tasks. But at least I know that the idea of using heuristics for long-term sub-goal assignment and learning-based policy for low-level sub-goal achievement is quite common in the RL field.

2. How the RL policy handles the collision and deadlock is unclear to me. What will happen if the agent (or two agents) choose the action(s) that will cause a collision? What will happen if there is a deadlock (e.g., two agents want to pass a narrow corridor)?

3. I am a little confused about what the RL policy can learn if the K is set to 2, which means the sub-goal is just two steps away from the current location. Are there many candidate paths to a sub-goal, which is just two steps away?

4. Lines 211-218: Since the agent doesn't know the future locations of other agents, how does the method count the "number of times the other agents were seen" in a future step? Does the method use a static heat map (for only the current step) to count that?

5. Lines 242 and 247: the symbol H was used twice with different meanings.

6. Figure 1 is not mentioned in the text.

7. Lines 175-176: "as the ratio of the episode length to the number of goals achieved" -> "as the number of goals achieved to the ratio of the episode length"?

8. Line 203: "node node"



**Questions:**

Can you provide more concrete qualitative examples to demonstrate that the learning-based policy is better at avoiding congestion or collision?

**Limitations:**

Yes

---

> ### Author Rebuttal · Authors · 2023-08-09
>
> ###  Weakness 1:
> Learnable low-level policies and heuristic sub-goal allocation procedures are commonplace in many hierarchical RL approaches tailored to single agent problems. However, such techniques are rarely explored in the context of multi-agent RL (MARL). Existing studies primarily demonstrate their results within simplistic environments, leaving ample room for further research in this area. In this context, we reference a fairly recent review paper [1] that highlights only two methods (PoEM [2] and HQMIX [3]) in the domain of decentralized partially observable multi-agent tasks. Among these, PoEM, a method closely related to ours, utilizes preexisting demonstrations to identify sub-goals, which poses a significant limitation. In contrast to our approach, all the methods we are aware of present their findings using scenarios with a small number of agents. Nonetheless, we intend to augment the section on related works by incorporating a comprehensive discussion of the advancements in the field of Hierarchical MARL.
>
> [1] Shubham Pateria, Budhitama Subagdja, Ah-Hwee Tan, and Chai Quek. 2021. Hierarchical Reinforcement Learning: A Comprehensive Survey. ACM Comput. Surv. 54, 5.
>
> [2] Miao Liu, Christopher Amato, Emily P. Anesta, J. Daniel Griffith, and Jonathan P. How. 2016. Learning for decentralized control of multiagent systems in large, partially-observable stochastic environments. In Proceedings of the 30th AAAI Conference on Artificial Intelligence (AAAI’16)
>
> [3] Hongyao Tang, Jianye Hao, Tangjie Lv, Yingfeng Chen, Zongzhang Zhang, Hangtian Jia, Chunxu Ren, Yan Zheng, Changjie Fan, and Li Wang. 2018. Hierarchical deep multiagent reinforcement learning. arxiv:1809.09332.
>
> ###  Weakness 2:
>
> The collision model adopted by us in this work is borrowed from Primal2 (our main learning-based competitor). When two or more agents decide to occupy the same cell at the next time step only one of them (decided by the environment) succeeds and the others stay where they were. When two agents wish to swap locations simultaneously (at the same time step), they both stay where they were.
>
> As for the deadlocks, they, indeed, can happen, and it is the responsibility of the policy to resolve them. The suggested hybrid policy (heuristic search + RL) handles the deadlock pattern mentioned by the reviewer - ‘two agents want to pass a narrow corridor’ - quite effectively. This can be seen in the animation, which is visible if one follows a link in Appendix A (this is a link to an anonymized repository, which contains a readme with an animation on the title page).
>
> ###  Weakness 3:
>
> We understand your confusion about the effect of setting K to 2, which implies that the sub-goal is only two steps away from the current location of the agent. Please note that Follower essentially aims to maximize rewards through the accomplishment of multiple sub-goals on the way to the (global) goal. Consider two illustrative scenarios:
>
> (1) The agent achieves a reward for reaching a sub-goal but subsequently encounters a conflict with another agent, impeding their further progress.
>
> (2) The agent allows another agent to pass and does not obtain an immediate reward for reaching sub-goal, but this action could actually lead to a more substantial reward later on (after successfully accomplishing several subsequent sub-goals).
>
> Our approach facilitates the agent's ability to learn the second type of behavior, thereby adapting its actions based on the potential for higher cumulative rewards in future. In terms of the training process, the agent learns using rewards discounted over full length trajectories (512 steps), which are divided into rollouts of size 8 for the RNN head.
>
> Thus, an agent's decision-making process isn't confined within the immediate two-step radius of its sub-goal. Instead, it extends towards a more distant time horizon. The rationale behind the selection of the hyperparameter value K=2, which emerged as the optimal choice in our hyperparameter sweep, is its ability to provide a dense reward signal.
>
>
> ###  Weakness 4:
>
> We assume that an agent does not know the future locations of the other agents. Thus, it uses the past observations to construct a heat map of the cost penalties.
>
> ### Weaknesses 5-8:
>
> Thank you, we will address these minor issues.
>
>
> ### Question 1:
>
> During training, each agent gains experience that in cases of congestion it might be preferable to yield to achieve a higher long-term reward (i.e. sacrifice short-time gain of achieving the local sub-goal in favour of achieving a sequence of sub-goals later on). Thus all agents (as the policy is shared) naturally learn to cope with deadlocks/congestions. If this learned policy is removed the performance significantly drops as confirmed by our experiments with Randomized A*, which is essentially Follower w/o learnable policy (see Fig. 5 in the original submission).

---

> > ### Comment · Reviewer_8Qnn · 2023-08-20
> > **Thank you!**
> >
> > Thanks to the authors for the detailed reply. Most of my concerns have been addressed.

---

### Author Rebuttal · Authors · 2023-08-09

We would like to express our gratitude to all the reviewers for their insightful reviews and comments. We appreciate that you found our work to be well-written and easy to follow. Additionally, we are pleased that you recognized the strength of our approach over both centralized planning and decentralized learnable methods, and your appreciation of the insights provided by our ablations.

We did our best to address the raised concerns (formulated both as weaknesses and direct questions to us). Here, we highlight three general points of our rebuttal.


1. Novelty. We agree with the reviewers that a combination of learnable low-level policies and heuristic sub-goal allocation procedures arises in many hierarchical RL approaches. However, they are mostly tailored to single-agent problems and are rarely explored in the context of multi-agent RL (MARL) and Multi-agent path finding (MAPF).  Moreover, existing studies primarily demonstrate their results within simplistic environments, leaving ample room for further research in this area.
 In the context of most hierarchical RL frameworks, an episode for a low-level agent typically revolves around a single sub-goal. Our method introduces a departure from this paradigm by allowing the Follower agent to pursue and accomplish multiple sub-goals along its trajectory. This creates opportunities for advanced decision-making policies, like preventing collisions with other agents via delaying instant rewards to achieve greater cumulative rewards. Additionally, a noteworthy innovation of our methodology is that a high-level subgoal generator tackles the challenge of conflict resolution in a long-term scenario by strategically distributing agents across the map (via the introduced cost-penalty heatmap).
2. Hyperparameters. On the one hand, it is true that the suggested hybrid method for solving (decentralized) Lifelong MAPF, i.e., Follower, requires setting various hyperparameters. On the other hand, hyperparameter selection is a necessary part of almost any learnable method. Moreover, in the path planning domains, even the non-learnable search-based methods often require setting their parameters. E.g., the state-of-the-art search-based LMAPF solver with whom we compare, i.e., RHCR, requires setting the re-planning frequency, planning horizon, etc. Having that said, we wish to emphasize that we tuned the hyperparameters of Follower only while training which was carried out only on the maze-like maps (from Primal2 paper).  Then, at the test time, we did not re-tune any hyperparameter, they all were left the same as when training had been finished. Subsequently, the results of Follower on out-of-distribution maps (i.e., on maps with topologies that it did not see during training) clearly demonstrate that Follower can generalize to unseen problem instances without requiring parameter re-tuning and thus a ‘hyperparameter concern’ is not crucial for Follower.
3. Empirical evaluation. For this rebuttal, we have conducted a range of additional experiments to address the concerns raised by the reviewers. I.e., we compared Follower to Primal2 on six more maps whose topology differs from the maps used for training Follower/Primal2 (see Fig. 1 in the attached file). Additionally, we performed further comparisons of Primal2 on randomly generated maps (with a high obstacle density of 30%), increasing the number of problem instances per map per agent count (see Fig. 2 in the attached file). We evaluated Follower, PICO, and Primal2 in the single-shot setup, measuring the success rate and the makespan (see Fig. 3 in the attached file). In all cases, Follower outperformed the competitors, providing additional evidence of the superiority of the suggested approach.

---

### Comment · Area_Chair_bqCP · 2023-08-15
**Please read and respond to authors' rebuttals**

Dear reviewers,

Thank you for your reviews. The authors have posted their rebuttal. If you have not yet done so, please read the rebuttal and the other reviews, and comment on whether the rebuttal has addressed your comments or concerns.

---

### Decision · Program_Chairs · 2023-09-21

**Decision:**

Reject

**Comment:**

This paper proposes to combine a heuristic-based path planner and an RL-learned obstacle avoidance policy for multi-agent path finding. All reviewers agree that the idea is simple yet effective, and the paper is well-written. Major concerns were also raised in the reviews, including novelty, hyperparameter tuning and performance. While the rebuttal has addressed most of these concerns, the concern about novelty still remains. Similar decomposition using long-horizon path planning (A*) and short-horizon local planner (RL) has been extensively studied in the literature, and has become a common practice in path planning/navigation. Admittedly, these work mostly focused on single agent scenarios. However, it seems relatively straightforward to apply the idea to the multi-agent domain. As a result, despite the emphasis on novelty in the rebuttal, the reviewers were not convinced. Their willingness to accept this paper is lukewarm. I agree with the reviewers' concern, and agree that the novelty of this paper has not reached the bar for NeurIPS.